# Evaluation of a Multiplex Real-Time PCR Assay for Detecting *Chlamydia trachomatis* in Vaginal Samples

**DOI:** 10.3390/diagnostics12051141

**Published:** 2022-05-04

**Authors:** Carole Kebbi-Beghdadi, Sebastien Aeby, David Baud, Gilbert Greub

**Affiliations:** 1Institute of Microbiology, Lausanne University Hospital and University of Lausanne, CH-1011 Lausanne, Switzerland; carole.kebbi-beghdadi@chuv.ch (C.K.-B.); sebastien.aeby@chuv.ch (S.A.); 2Materno-Fetal and Obstetrics Research Unit, Department Woman-Mother-Child, Lausanne University Hospital, CH-1011 Lausanne, Switzerland; david.baud@chuv.ch

**Keywords:** *Chlamydia trachomatis*, multiplex PCR, urogenital infections, pre-term birth, prevalence

## Abstract

*Chlamydia trachomatis* is an important cause of sexually transmitted infections (STI) in Western countries. It is often asymptomatic, and thus, left untreated, and can have severe negative consequences, such as tubal infertility or adverse pregnancy outcomes. Other sexually transmitted microorganisms, such as *Neisseria gonorrhoeae* and *Trichomonas vaginalis*, as well as normal residents of the vaginal flora, such as genital mycoplasmas, also negatively impact human sexual and reproductive health. We evaluated the reliability of the Seegene Allplex STI Essential Assay for *C. trachomatis* detection using the real-time qPCR routinely used in our diagnostic laboratories as the gold standard. The Seegene assay displayed a sensitivity of 97.8% and a specificity of 98.9%. As this assay can also detect six other urogenital pathogens, we applied it to 404 samples from women who attended Lausanne University Maternity Hospital and obtained the following prevalence rates: 2.5% for *C. trachomatis*, 3.5% for *Mycoplasma hominis*, 6.3% for *Ureaplasma* *urealyticum*, and 27.7% for *U**reaplasma* *parvum*. Two samples were positive for *Trichomonas vaginalis*, and one sample was positive for *Mycoplasma genitalium*. Bacterial vaginosis was present in 4.5% of the cases and was strongly associated with *M. hominis*. Finally, we confirmed the association between *C. trachomatis* infection and pre-term birth (*p* = 0.03) but could not detect any association of this condition with other urogenital pathogens (Mycoplasma/Ureaplasma). In conclusion, given its high sensitivity and specificity for *C. trachomatis* DNA detection as well as its multiplex format, which simultaneously provides results for six other urogenital pathogens, the Seegene Allplex™ STI Essential Assay represents an appealing diagnostic tool in modern microbiology laboratories.

## 1. Introduction

*Chlamydia trachomatis* is the leading cause of sexually transmitted bacterial infections worldwide with over 127 million cases estimated in 2016, according to the most recent data from the World Health Organization (WHO) [1].

In symptomatic cases, *C. trachomatis* infection causes several urogenital syndromes, including urethritis, epididymitis, prostatitis, and cervicitis [2]. However, 70–80% of *C. trachomatis* infections in women and 50% in men are asymptomatic, which probably explains the relatively high incidence of this sexually transmitted infection (STI) even in Western countries with advanced medical systems and public health programs [1].

In some women, untreated *C. trachomatis* infections may ascend to the upper genital tract and become chronic, leading to long-term sequelae, such as pelvic inflammatory disease (PID), ectopic pregnancy, and tubal infertility (TFI), or inducing adverse pregnancy outcomes [2,3,4]. Infections are also associated with an increased risk of cervical cancer and HIV acquisition and transmission [5,6].

No effective vaccine against *C. trachomatis* has been developed so far, and routine screening of sexually active young women is currently the only method to prevent complications of asymptomatic infections [7].

*Neisseria gonorrhoeae* and *Trichomonas vaginalis* are also important sexually transmitted pathogens that negatively affect human sexual and reproductive health [8]. Finally, genital mycoplasmas, such as *Mycoplasma genitalium*, *Mycoplasma hominis*, *U**reaplasma urealyticum,* and *U**reaplasma parvum*, can be normal residents of the vaginal flora but have also been associated with adverse pregnancy outcomes, such as preterm labor and premature rupture of membranes [9,10,11,12]. Their role is complicated by the presence or absence of bacterial vaginosis [9].

All these urogenital microorganisms can be detected by Nucleic Acid Amplification Tests (NAATs), which display high sensitivity and high specificity and can be performed with non-invasive clinical samples, such as urine or vaginal swabs [13]. Furthermore, several companies have now developed multiplex real-time PCR assays that can detect multiple microorganisms in a single assay.

In this study, we evaluated the power of the Seegene Allplex™ STI Essential Assay, which can simultaneously detect *C*. *trachomatis*, *N*. *gonorrhoeae*, *T*. *vaginalis*, *M*. *genitalium*, *M*. *hominis*, *U*. *urealyticum,* and *U*. *parvum*, as a diagnostic tool. We first evaluated the reliability of this assay for *C. trachomatis* detection compared with the real-time qPCR routinely used in our diagnostic laboratories (used as the gold standard). Second, we used two Seegene assays to determine the prevalence of the seven microorganisms mentioned above and bacterial vaginosis in a cohort of 404 women who attended Lausanne University Maternity Hospital. Finally, to assess the role of these pathogens in pre-term birth (PTB), we compared their prevalence in a group of women who gave birth prematurely with that in women with uneventful term pregnancies.

## 2. Methods

### 2.1. Samples

Patient samples were collected at the obstetrical ward of Lausanne University Maternity Hospital during a previous study [14]. The complete cohort included 404 samples; however, different subsets of this cohort were used for the different analyses listed below.
To assess the reliability of the Seegene Allplex™ STI Essential Assay for the detection of *C. trachomatis*, we used 269 samples that were previously tested with the *C. trachomatis*-specific real-time PCR routinely used in our diagnostic laboratories [15]. Since among these 269 samples, only 1 was positive for *C. trachomatis*, 45 samples collected at Lausanne University Hospital between May and July 2021 and assigned as *C. trachomatis*-positive using this same PCR [15] were added to the group to improve our evaluation of the assay.To calculate the prevalence of urogenital microorganisms and bacterial vaginosis, we used the complete cohort, i.e., 404 samples. The median age of the patients was 30 +/− 5.5.To assess the association of urogenital microorganisms and bacterial vaginosis with pre-term birth, we analyzed 217 samples from women who attended a labor ward with uneventful term pregnancies and no history of miscarriages, stillbirths, or pre-term labor and 97 samples from women who spontaneously delivered before 37 weeks of gestation (PTB). The median age of the patients was 29 +/− 4.2 in the control group and 31 +/− 5.9 in the PTB group (*p* = 0.445).

### 2.2. DNA Extraction and Storage

DNA from the 404 samples of the cohort had been extracted, following the manufacturer’s instructions, with the QIAamp DNA Mini Kit (Qiagen, Hilden, Germany) at the time of the previous study [14] and had been kept frozen since then. DNA from the 45 more recent samples was extracted with the same kit.

### 2.3. Seegene Multiplex Real-Time PCRs

All samples from the cohort (*n* = 404) were tested using the multiplex real-time PCR Allplex™ STI Essential Assay (Seegene, Seoul, Korea), which can simultaneously detect *C*. *trachomatis*, *N*. *gonorrhoeae*, *T*. *vaginalis*, *M*. *genitalium*, *M*. *hominis*, *U*. *urealyticum,* and *U*. *parvum* in a single tube using dual priming oligonucleotide (DPO™) and multiple detection temperature (MuDT™) technologies. They were also tested with the Allplex™ Bacterial Vaginosis Assay (Seegene, Seoul, Korea), a multiplex real-time PCR assay that simultaneously detects 7 bacteria species associated with vaginosis (*Atopobium vaginae*, bacterial vaginosis-associated bacteria 2, *Bacteroides fragilis*, *Gardnerella vaginalis*, *Lactobacillus* spp., *Megasphaera* Type 1, and *Mobiluncus* spp.).

The 45 *C. trachomatis*-positive samples were only analyzed with the multiplex real-time PCR Allplex™ STI Essential Assay (Seegene, Seoul, Korea).

### 2.4. In House-Developed Real-Time qPCRs

A total of 269 samples from the cohort and 45 samples collected later (see above) were tested with a *C. trachomatis*-specific real-time qPCR targeting the *C. trachomatis* cryptic plasmid [15]. This qPCR, developed in-house, is routinely used in our diagnostic laboratories for *C. trachomatis* detection. It amplifies DNA during 45 cycles.

The 45 above-mentioned samples were also tested with a pan-*Chlamydiales* real-time qPCR that amplifies a region of the 16SrRNA gene conserved in all *Chlamydiales* bacteria [16]. This qPCR also amplifies DNA during 45 cycles.

### 2.5. Statistical Analyses

Patient ages were compared using an unpaired *t*-test. Categorical variables were compared using Fisher’s exact test.

## 3. Results and Discussion

### 3.1. Reliability of the Seegene Allplex™ STI Essential Assay for Detection of C. trachomatis

In order to evaluate the reliability of the Seegene Allplex™ STI Essential Assay for the detection of *C. trachomatis*, we analyzed with this assay 269 DNA samples collected from women who attended Lausanne University Maternity Hospital and manually extracted from vaginal swabs. These same samples were previously analyzed with a *C. trachomatis*-specific real-time qPCR targeting the cryptic plasmid, which is routinely used in our diagnostic laboratories [14,15]. In addition, the Seegene Allplex™ STI Essential Assay was also applied to 45 samples from our diagnostic laboratories previously determined as *C. trachomatis*-positive using the real-time qPCR mentioned above.

Altogether, of these 314 samples, 46 were positive in the *C. trachomatis* qPCR, while the Seegene Allplex™ STI Essential Assay detected 48 positive samples (Table 1). Only one sample classified as positive with the *C. trachomatis* real-time qPCR (Ct value: 38.1) was not detected as positive with the Seegene assay (1 false negative). However, this very low Ct value corresponds to less than 100 DNA copies/mL and is very close to the limit of detection of the qPCR assay [15]. Moreover, the Seegene Allplex™ STI Essential Assay detected three positive samples that were classified as negative by the classical qPCR (3 false positives).

Thus, when considering the *C. trachomatis*-specific real-time qPCR as the gold standard, the Seegene Allplex™ STI Essential Assay displayed a sensitivity of 97.8% and a specificity of 98.8%.

In addition, the 45 samples from our diagnostic laboratories that were positive in the *C. trachomatis* qPCR were also analyzed with the pan-*Chlamydiales* real-time qPCR. This qPCR, developed in our research group, targets a region of the 16SrRNA gene conserved in all bacteria belonging to the *Chlamydiales* order, including *C. trachomatis* [16]. Three samples positive in the *C. trachomatis* qPCR and the Seegene assay were negative in the pan-*Chlamydiales* qPCR, which confirms that this last test is less sensitive for the detection of *C. trachomatis* than the *C. trachomatis*-specific qPCR. This difference in sensitivity can be explained by the size of the amplified fragment, which is notably longer in the case of the pan-*Chlamydiales* qPCR (207–215 bp) compared with the *C. trachomatis*-specific qPCR (71 bp) [15,16].

The Seegene false-negative sample was positive in the pan-*Chlamydiales* qPCR although with a very high Ct value (39.90). To summarize, among 45 samples positive in the *C. trachomatis*-specific real-time qPCR, the Seegene assay failed to detect one (2.22%), and the pan-*Chlamydiales* failed to detect three (6.66%). As expected, this real-time qPCR is the least sensitive of the three NAATs used in this study.

### 3.2. Prevalence of Urogenital Microorganisms and Bacterial Vaginosis

Four hundred four samples from women who attended Lausanne University Maternity Hospital were analyzed with the Seegene Allplex™ STI Essential Assay. Seven samples were considered invalid by the analysis software, indicating some issues with the internal validation control of the assay. According to the manufacturer’s package insert, such invalid outcomes are typically due to inadequate specimen collection or the presence of inhibitors. However, these seven samples had been previously tested with the *C. trachomatis* specific real-time qPCR and were negative, with no evidence of the presence of inhibitors.

Among these 404 samples, 10 were positive for *C. trachomatis*, and 5 of these 10 were also positive for one or two other microorganisms detected by the same assay (see Appendix A). Our results thus indicate a prevalence of *C. trachomatis* infection of 2.5% in this group of women (Table 2). Interestingly, another study performed in western Switzerland revealed a slightly higher prevalence of *C. trachomatis* infection (5.9%) among sexually active young women under the age of 30, a value comparable to the one reported for European countries [17,18]. However, the populations of women enrolled in the two Swiss studies are not comparable, particularly regarding their age. Indeed, the median age of positive patients in the study by Bally et al. [17] was 20.3, whereas the median age of positive patients was 26 in our study. The results from these two studies thus suggest that the prevalence of *C. trachomatis* infection in young women in Switzerland is similar to that described in other countries and decreases with age [19,20,21].

In addition to *C. trachomatis*, the Seegene Allplex™ STI Essential Assay detects six other microorganisms, among which are *N. gonorrhoeae* and *T. vaginalis*, two important causes of sexually transmitted infections worldwide. We did not detect any *N. gonorrhoeae* infection among the 397 valid samples tested, and only two samples were positive for *T. vaginalis* (prevalence: 0.5%) (Table 2). These low values of prevalence are in accordance with what has been observed in other studies [22,23].

Finally, as shown in Table 2, the prevalence of genital mycoplasmas in our cohort was in agreement with what has been reported in previously published studies [9,24]. *M. genitalium* and *M. hominis* were detected in 0.3% and 3.5% of the samples, respectively, while *U*. *urealyticum* and *U*. *parvum* were detected, respectively, in 6.3% and 27.7% of the samples.

For the same group of samples, we also performed the Seegene Allplex™ Bacterial Vaginosis Assay and detected 18 positive cases (4.5%). All were positive for *Gardnerella vaginalis*, two samples were also positive for *Mobiluncus* spp., and 12 samples were positive for *Atopobium vaginae* (see Appendix A). Thirteen of the eighteen positive samples (72.2%) were associated with one or two of the seven microorganisms assessed with the Seegene Allplex™ STI Essential Assay (see Appendix A). Similar to what is described in the literature [9,25], *M. hominis* seemed to be strongly associated with bacteria, labeled in the Seegene assay as “bacterial vaginosis”, since 33.3% (1/3) of the samples positive only for this microorganism were also positive for “bacterial vaginosis” bacteria. *U*. *urealyticum* and *U*. *parvum* were also associated with “bacterial vaginosis” bacteria, but to a lesser extent (6.6% (1/15) and 6.5% (6/93)) of the *Ureaplasma* positive cases, respectively) [25]. Samples positive only for *C. trachomatis*, *T. vaginalis,* or *M. genitalium* were negative for “bacterial vaginosis” bacteria, as previously reported in other studies [25,26].

### 3.3. Association of C. trachomatis and Genital Mycoplasmas with Pre-Term Birth (PTB)

We then wished to determine if one of the microorganisms detected by the Seegene Allplex™ STI Essential Assay or the Bacterial Vaginosis Assay could be implicated in pre-term birth (PTB). For this purpose, we compared the prevalence of these microorganisms in a group of 97 women who spontaneously delivered before 37 weeks of gestation (PTB group) and in 217 women who attended a labor ward with uneventful term pregnancies and no history of miscarriages, stillbirths, or pre-term labor (control group) (Table 3) [14,27]. Our results indicated that *C. trachomatis* infection was significantly associated with pre-term birth (*p* = 0.03), as already described by others [28] and by Baud et al. in a previous study performed on the same samples [14]. No significant association with pre-term birth was observed for *M. hominis*, *U. urealyticum*, or *U. parvum* in the present study, although a role for these microorganisms in pre-term birth has been demonstrated previously [9,10]. Pre-term birth was also not associated with bacterial vaginosis. For statistical reasons, pathogens that were also detected with the Seegene Allplex™ STI Essential Assay but exhibited a prevalence of <1% (*N. gonorrhoeae*, *T. vaginalis,* and *M. genitalium)* could not be investigated regarding their possible role in pre-term birth and are thus not listed in Table 3.

## 4. Conclusions

Using a different NAAT from that in the study by Baud et al. [14], we confirmed a prevalence of 2.5% for *C. trachomatis* in a population of women, with a median age of 30, who attended Lausanne University Maternity Hospital. We also confirmed a statistically significant association between this pathogenic microorganism and pre-term birth.

Furthermore, we demonstrated the reliability of the Seegene Allplex™ STI Essential Assay in detecting *C. trachomatis* in DNA extracted from vaginal swabs. The sensitivity and specificity of this assay were about 98% and 99% when compared with the *C. trachomatis* real-time qPCR routinely used in our diagnostic laboratories. Given that this assay can simultaneously detect six other microorganisms, it is of high interest for the diagnosis of urogenital and sexually transmitted infections.

## Figures and Tables

**Table 1 diagnostics-12-01141-t001:** Comparison of the performance of the Seegene Allplex™ STI Essential Assay and of our routine *C. trachomatis* qPCR. Please note the excellent congruent results obtained, with an overall agreement of 98.7% (310/314).

		Seegene Allplex™ STI Essential Assay	
		Positive	Negative
qPCR	positive	45	1
*C. trachomatis*	negative	3	265

**Table 2 diagnostics-12-01141-t002:** Number of positive cases and prevalence of *C. trachomatis*, *N. gonorrhoeae*, *T. vaginalis*, *M. hominis*, *M. genitalium*, *U. urealyticum*, *U. parvum*, and bacterial vaginosis among 397 women who attended Lausanne University Maternity Hospital (median age = 30 +/− 5.5).

	Positive Cases	Prevalence
*Chlamydia trachomatis*	10	2.5%
*Neisseria gonorrhoeae*	0	0.0%
*Trichomonas vaginalis*	2	0.5%
*Mycoplasma hominis*	14	3.5%
*Mycoplasma genitalium*	1	0.3%
*Ureaplasma urealyticum*	25	6.3%
*Ureaplasma parvum*	110	27.7%
Bacterial vaginosis *	18	4.5%

* PCR positive for *G. vaginalis* +/− other bacteria such as *Mobiluncus* spp. and *A. vaginae.*

**Table 3 diagnostics-12-01141-t003:** Number of positive cases and prevalence of *C. trachomatis*, *M. hominis*, *U. urealyticum*, *U. parvum,* and bacterial vaginosis in women with uneventful pregnancy (control group, *n* = 217) and in women who spontaneously delivered before 37 weeks of gestation (PTB group, *n* = 97).

	Controls (*n* = 217)	PTB (*n* = 97)	*p*-Value
*Chlamydia trachomatis*	2 (0.92%)	5 (5.15%)	0.031
*Mycoplasma hominis*	6 (2.76%)	5 (5.15%)	0.324
*Ureaplasma urealyticum*	12 (5.53%)	10 (10.31%)	0.151
*Ureaplasma parvum*	54 (24.88%)	33 (34.02%)	0.103
Bacterial vaginosis *	8 (3.69%)	4 (4.12%)	>0.999

* PCR-positive for *G. vaginalis* +/− other bacteria, such as *Mobiluncus* spp. and *A. vaginae*.

## Data Availability

The datasets generated and/or analyzed during the current study are available from the corresponding author on reasonable request.

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
