# Peer review of "Evaluation of a Multiplex Real-Time PCR Assay for Detecting Chlamydia trachomatis in Vaginal Samples"

_diagnostics, 2022, doi:10.3390/diagnostics12051141_

Round 1

Reviewer 1 Report

The authors present an interesting study where they have evaluated the performance of a multiplex commercial assay for the detection of various microorganism in vaginal swabs from clinical patients.

The assay used in the study is the Seegene Allplex™ STI Essential Assay which has the capacity to detect Chlamydia trachomatis and six other sexual transmitted infectious microorganisms. The study focused on the detection of C. trachomatis, concluding the sensitivity and specificity of the assay was 97.8% and 98.9% respectively for this pathogen. The study was extended to a clinical investigation where a significant association between preterm births and a positive result C. trachomatis was identified.

Overall the study is well written and the conclusions are supported by the presented data.

Suggestions and comments.

Line 2 I would suggest the authors revise the title of their manuscript. There is limited information on the reliability of the evaluated assay. Perhaps something along the lines of:

Evaluation of a multiplex qPCR assay for detecting sexually transmitted infections in maternal patients

Line 39 suggest revision “infection (STI) in western countries”

Line 74 I am assuming the human ethics approve is quote in Baud et al. [14], I would suggest either stating this here or quote the relevant approval.

Line 91 Was the input amount of DNA into the qPCR standardised for each sample?

Line 130 I would suggest replacing the symbols “+” and “-“ with “positive” and “negative”, respectively. There is ample room in the table and use of the words provides clarity.

Line 147 suggest revision “with a very high Ct value (39.90).”

Of course, this is indicative of low template concentration. It may be worth considering adding the number of cycles to the methods.

Lines 163-167 The authors should carefully review this text. While 47% of women in the quoted study, “fit” within the authors proposed hypothesis, that younger women are more likely to test positive, it also means that 53% of women did not. Thus comparing a percentage from that study to the median age in the current study could be misleading. Is it possible to report the data up from the Bally et al. study so that prevalences in younger women defined and old women are separated with the groupings defined median ages?

I would request the authors review this text to ensure it reflects the factual data in the quote studies. Some of the text, e.g. Line 166 “which implicates (sic implies) that these patients have a (sic) stable” could be interpreted as reinforcing gender biases/stereotypes in scientific reporting. If the studies do not specifically report ages and/or number of sexual partners as recorded from their cohorts during data collection, these should not be implied/assumed in the current study.

Line 167 suggest revision “Bally et al. [17].

Line 177 Here and elsewhere in the manuscript genus names should be abbreviated after first use by scientific convention.

Lines 216-219 this text appears to be written as a footnote to the table. If so, it should be designated as such using relevant superscripts. If the text is supposed to describe the results presented in the table, the table should be referenced correctly.

Line 221 suggest revision “by Baud et al. [27] we”

Line 223 suggest revision “we also identified a statistically significant”

Reviewer 2 Report

The manuscript entitled: “Chlamydia Trachomatis PCR: High Reliability of the Seegene 2 Multiplex Assay” addresses an interesting topic. All in all, on my view point, the study have a merit to be published, however several points need to be addressed before any proceeding.

  • The discussion section has been omitted and should be added. Of course, I think the authors wrote the discussion and the results together, which in this case, must be improved.
  • P value should be written in italic.
  • It is better to add patients' demographic information and their analysis.
  • Ethical approval of the research is required.
  • Statistical analysis should be performed to compare the methods mentioned in the study.

Reviewer 3 Report

In this study, Kebbi-Beghdadi and co-workers compare the performance of the Seegen Allplex STI Essential Assay and an in-house PCR for diagnosis of Trichomonas vaginalis. In addition, a sample collection was screened with the Seegene assay for additional urogenital microorganisms and the association of vaginosis-inducing pathogens with pre-term birth. Since syndromic-based panels are a valuable tool in molecular diagnostics, the performance analysis of the Seegene assay using clinical samples is of high importance. The manuscript is well-written and has merit for publication and only minor points need to be addressed. Heading: please change to “…Chlamydia trachomatis…” Line 14-18: please rephrase this sentence Line 20: please consider to change the term “Maternity of Lausanne University Hospital” throughout the entire manuscript to “Lausanne University Maternity Hospital” as used in line 116. Samples: please comment on the storage of samples. Are they extracted DNA and stored frozen since 2006-2009? Or are they residual original samples and the DNA was extracted as described by the Qiagen kit? Line 93: n=404 is a contradiction to line 76. Are 404 or 269/314 samples used for sensitivity testing of the Seegene assay? See also line 122. Line 99: please change to “…bacteria species…” Method section: please comment on the statistical methods. Line 112. Please change to “Results and discussion” Line 113: please change to “…Chlamydia trachomatis” Line 119-122: repetition to the Methods section Line 151: please rephrase this header Line 154: what is the context of the SARS-CoV-2 pandemic for this manuscript? Line 155: Are these invalid samples tested before by PCR? If so, is there an evidence for presence of inhibitors? Line 158: please consider to show which other pathogens are present in the C. trachomatis positive samples. Line 187-189: did these patients had a diagnosis for vaginoses? Line 189: see comment to line 158 and consider presenting which microorganisms were found in the same sample. Line 197: please delete the full stop Line 204: please change to “…with pre-term birth (p=0.03).” and delete line 216. Line 209: please change to “Because the prevalence…M. genitalium was lower than…” Line 217-219: please change to “For statistical reasons pathogens also detected with the Seegene Allplex STI Essential Assay exhibiting a prevalence…” References: please carefully check the reference section for the correct style (e.g. journal names, italics, …) END OF COMMENTS

Reviewer 4 Report

The study entitled “Chlamydia Trachomatis PCR: High Reliability of the Seegene Multiplex Assay” evaluated the sensitivity and specificity of the Seegene Allplex™ STI Essential Assay for C. trachomatis detection. Moreover, the role of the seven pathogens detected by Seegene Allplex™ STI Essential Assay in pre-term birth (PTB) was also assessed.

Below, I pinpoint a few suggestions and comments that must be addressed:

  • Line 122: “Altogether, on these 314 samples, 46 returned positive with the qPCR…” shouldn´t we read ‘45’ instead of ‘46’? For instance: Altogether, on these 314 samples, 45 returned positive with the qPCR. 45 is what is on Table 1.
  • Lines 156-158: I suggest: “According to the manufacturer’s package insert, such invalid outcomes are typically due to inadequate specimen collection or presence of inhibitors.”
  • Line 167: “enrolled in the study of Bally et al.” The reference number is missing (must be inserted after Bally et al.).
  • In Table 2: Correct for “Trichomonas” and not “Trachomonas”
  • Table 2 and Table 3: Bacterial vaginosis is reported but there is no association to the specific microorganism. It would be of interest to associate the bacterial vaginosis to the agent (one of the seven) in each case. This part and these results have to be better presented and discussed.
  • Line 221: Reference number is missing.

Round 2

Reviewer 2 Report

Accept